# A Comparative Study on Supervised Machine Learning Algorithms for Copper Recovery Quality Prediction in a Leaching Process

**DOI:** 10.3390/s21062119

**Published:** 2021-03-17

**Authors:** Victor Flores, Claudio Leiva

**Affiliations:** 1Department of Computer and Systems Engineering, Universidad Católica del Norte, Antofagasta 1270709, Chile; 2Department of Chemical Engineering, Universidad Católica del Norte, Antofagasta 1270709, Chile; cleiva01@ucn.cl

**Keywords:** data analysis, artificial intelligence, machine learning, knowledge engineering, computers and information processing, data analysis, data processing

## Abstract

The copper mining industry is increasingly using artificial intelligence methods to improve copper production processes. Recent studies reveal the use of algorithms, such as Artificial Neural Network, Support Vector Machine, and Random Forest, among others, to develop models for predicting product quality. Other studies compare the predictive models developed with these machine learning algorithms in the mining industry as a whole. However, not many copper mining studies published compare the results of machine learning techniques for copper recovery prediction. This study makes a detailed comparison between three models for predicting copper recovery by leaching, using four datasets resulting from mining operations in Northern Chile. The algorithms used for developing the models were Random Forest, Support Vector Machine, and Artificial Neural Network. To validate these models, four indicators or values of merit were used: accuracy (acc), precision (p), recall (r), and Matthew’s correlation coefficient (mcc). This paper describes the dataset preparation and the refinement of the threshold values used for the predictive variable most influential on the class (the copper recovery). Results show both a precision over 98.50% and also the model with the best behavior between the predicted and the real values. Finally, the obtained models have the following mean values: acc = 0.943, p = 88.47, r = 0.995, and mcc = 0.232. These values are highly competitive when compared with those obtained in similar studies using other approaches in the context.

## 1. Introduction

At present, the copper mining industry in Chile and the world is undergoing big changes owing to the use of modern techniques such as predictive models based on Artificial Intelligence [1,2,3]. An example of them is the use of data mining methods to study or predict copper recovery or mineral prospects [4].

These methods relate to data-driven techniques used in several engineering areas, showing good results in most cases [5]. Techniques such as machine-learning algorithms have been recently used in the copper production industry to, for instance, reduce the cost of leaching methods aiming to improve both processes and results [6,7].

A proper amount of data related to processes and results are necessary for these algorithms to work, but not all machine-learning algorithms work properly in these domains. This is because the copper production industry neither provides good results in all domains nor works on relevant datasets concerning the suitability of data and accuracy rates [8,9].

There are well-known methods used for producing metallic copper via hydrometallurgy in the copper industry: dynamic heap leaching, solvent extraction, and electro-winning. The ultimate goal of these processes is to produce the most amount of copper, saving resources and being the least aggressive for the environment. However, the quality of copper recovery is difficult to predict with this method due to the big number of process parameters (e.g., metals present, granulometry of the material to be leached, and soluble copper grade, among others) and other aspects, such as performance fluctuations depending on time due to complex irrigation systems and product validation.

Recent studies such as [10] report predictive copper recovery models with 95% accuracy, utilizing artificial neural networks and parameters widely used in industry, such as “Monoclass granulometry”, “Irrigation rates”, “Total acid added”, “Pile high”, “Total copper grade”, “CO_3_ grade”, “Leaching ratio”, “Operation day”, and “Soluble copper grade stacked”.

### Advantages and Disadvantages of Data-Driven Approaches in Copper Mining

Recent studies highlight an increasing interest in using machine-learning algorithms to develop predictive models for the mining industry [3,8].

Recently, a big number of studies on data-driven approaches have been published. They deal with the use of predictive model techniques for metallic copper recovery processes. For example, the authors of [3] reported artificial intelligence methods used for developing predictive copper recovery models. For this kind of works, Gradient Boosted Trees (GBT), Random Forest (RF), Support Vector Machine (SVM), and Artificial Neural Network (ANN) are some of the most frequently used methods. Research on predictive copper models or copper recovery models considers this issue from different perspectives. Recent works reported models using different data mining techniques [1,4,10,11].

There are recent studies that compared the results of predictive models used in mining. For example, the authors of [12] compared ANN, wavelet neural network (WNN), and SVM to classify the mapping of potential copper points in an Iranian copper mine. The best result was obtained with ANN. Other recent studies compared predictive model performance in other industries, as detailed in [13], where a systematic literature review related to this domain was detailed.

For example, the authors of [14] presented a study based on an ANN algorithm and focused on the use of fly ash as supplementary cementitious material, the authors of [15] described an interesting work using ANN, SVM and RF to generate predictive models for the behavior of organic solvent nanofiltration membranes used in the chemical industry (particularly in the pharmaceutical sector), and the authors of [16] compared the use of ANN, SVM, and RF in the formation of geological reservoirs, while the authors of [17] compared ANN and RF in rock drilling and blasting in a mining company. In this study, the authors concluded that the ANN-based model showed the best performance.

The authors of [10] reported results from the generation of predictive models for copper recovery by leaching. In this study three mathematical models (lineal, quadratic, and a cubic), and an ANN model were developed and the results of the prediction for all of them were compared. The best predictive values were obtained with ANN (97% model accuracy).

More recently, the authors of [3] reported results from the generation of predictive models for copper recovery by leaching, using a different dataset with respect to [10]. In this study, a predictive model for copper recovery using RF was developed and validated with data collected at the plant and in a controlled environment in the laboratory. The resulting model showed the relevance of the predictive variables over the class (copper recovery), showing 95% accuracy.

To develop models using data-driven techniques, the stages generally grouped together in [1,2,3,8,9,10,11,12] are as follow: (1) data acquisition and curation and data verification, (2) model development, (3) validation of models developed and interpretation of results, and, depending on the study objectives, and stage (4) consists in using the model to generate results, e.g., by prediction.

The models developed with data-driven techniques, as compared with those developed with other techniques, have the following advantages: they are more intuitive for interpretation when they are made by mining exploitation experts. However, the non-specialists in artificial intelligence techniques can help identify non-evident relations between some aspects of the context to improve performance and reduce necessary resources and processing time, among others.

The following disadvantages can be mentioned: to use a supervised learning technique, it is necessary to have data, good data, and also previous knowledge to generate a good training set. The combination of data and experience is not quite frequent in mining since experimentation may be too expensive. In addition, the time of one or several experts for participating in tasks such as data cleaning and model training may be too costly and even unviable for mining companies.

Furthermore, it is not feasible to know a priori the most appropriate data mining method meeting certain characteristics of a mining exploitation since a big number of parameters come into play, some of which cannot be measured in operation or may vary, for example, mineralogy.

A comparison between the models already developed must be made to evaluate the most appropriate data mining methods and generate another model. For example, the literature reports comparisons between models developed with Bayesian networks and ANN. The authors of [9,16] reported some examples but in other industrial contexts. However, the literature does not show studies on this comparison for copper extraction by leaching.

This study aims to compare the performance of three supervised machine learning algorithms used to develop predictive models of copper recovery by leaching. Here, an assessment using cross-validation accuracy among three algorithms is presented: Artificial Neural Networks (ANN), Support Vector Machines (SVM), and RF. It aims to compare the results and determine the one contributing with the most knowledge to the leaching process. The question of what can be learned, discovered, and predicted with each model will be addressed and the quality measures of each model to compare their performance will be determined. All this follows previous works [3,18].

The paper is organized as follows: Section 2 describes the 1-year copper leaching process studied by comparing three models to predict copper heap leaching results. Section 2 also compares traditional and machine-learning methods to predict copper recovery by leaching. Section 3 describes artificial intelligence methods used for developing predictive models, particularly, machine learning, the methodology used and its stages, and the experimental design of the models devised. Section 4 describes the implementation of the models and their validation using performance measures and cross-validation. Section 5 deals with the analysis and discussion of results, along with future research lines. Finally, acknowledgments and bibliography are included.

## 2. Related Works and Context

As described by the authors of [10], the Franke mining company uses three processes well-known in the copper mining industry. One of them being of interest in this study is leaching in dynamic cells. In heap leach piles, high mineral variability and irrigation conditions used in the leaching process increases the system entropy, a condition that causes excess uncertainty (close to ±20%) with respect to the copper recovery resulting at the end of the operational cycle.

This usually involves poor planning of the irrigation module start-up, agglomerate and irrigation conditions, drainage distribution, inventories of poor solutions, Pregnant Leaching Solution (PLS) flow to the Solvent Extraction (SX) plant, and copper deposited and extracted at the electrowinning (EW) plant. The leaching process is similar in most mining companies. Principal steps of the mining leaching process are represented in Figure 1.

The ore is obtained from the open pit mine by extracting copper oxides mostly composed of atacamite. Then, it is sent to the primary crushing section by means of belts. The ore mixture obtained from the primary crushing is classified according to granulometry and then, sent to either a stockpile, or a secondary or tertiary crusher. If the material meets the granulometry of the company, it is sent to stockpile 1 (defined as coarse stockpile) and later to the secondary crusher.

The pile-leaching of copper is a percolation process that operates above the ground, as illustrated in Figure 1. The oxide copper ore is piled up on leach pads which have a slope of about 3° and a rubber lining that seals the ground beneath. These heaps are 3 m high and 72 m long, with a base area of 2880 m^2^, covering 120,000 tons of ore.

The grain classification is checked again, and the ore is sent to the tertiary crusher, if necessary. Finally, it goes to stockpile 2 (fines), where it is stored to finally enter the agglomeration process. This process takes place in two agglomeration tanks with the same dimensions. Here, the ore gets in contact with a sulfuric acid solution and water. The objective of this process is to form uniform agglomerates of a proper size with the fine material. In addition, it contributes to copper sulfation due to its contact with acid solutions.

At the end of this process, the material is sent to the dry area. Here, stock-piling ends. A radial stockpiler forms a dynamic stockpile used for leaching. It has a maximum of 12 slots, each of them formed by eight modules. Each module has certain dimensions and operates with 10,000–15,000 tons of agglomerate.

The process to produce copper ends with the drainage of the solution going to PLS. The copper in the PLS solution has an average concentration which is assessed at different times during the stockpile operation.

These measurements allow determining the quality of the leached copper and also help to validate both the planning process of the leaching stockpile life cycle and also how well each part of the process is designed for the stockpile to operate. At this moment, two partial results are obtained: PLS and Intermediate Liquid Solution (ILS) solutions. The PLS solution is used in the solvent extraction stage, while the ILS solution is used for leaching new modules. Stockpile exploitation is conducted during a period of time called exploitation cycle. These may last 30–45 days in which the tasks described above take place. The goal in these processes is to achieve the highest copper production by saving resources with the lowest possible environmental impact.

In this regard, during the leaching stockpile exploitation, the company collected copper recovery data and also data in a controlled environment with stockpile material, although piled in columns and processed in the laboratory. In this study and similar to other studies [3,10], the dataset directly obtained from the stockpile is called operational data, while those obtained from the columns in the laboratory are called column data. Further details about the datasets, how they were obtained, and the pre-process to be used in the models are described in the next section.

### 2.1. Traditional and Machine Learning Processes to Predict Copper Recovery by Leaching

One of the most common difficulties companies face when using the leaching technique to recover copper is the lack of historical data for planning the stockpile exploitation and forecasting or planning the operation [3]. Some of the simplest prediction techniques consists in generating or using random variables utilizing iterative simulation and then, conduct goodness-of-fit tests [6].

However, these techniques usually result in high interdependence indexes, possible errors, and subjective interpretation of significance [19]. Therefore, other statistical and mathematical techniques such as time series regression models are used [20]. These kinds of techniques have been used in copper mining industry for several decades, but the optimization of the results in this domain is possible using machine learning techniques as shown in previous works [3,10,17,20].

For example, the authors of [21] developed a detailed and accurate simulation model capturing factors that affect air due to copper production and the handling of materials used in the productive activity of an Australian mine. This study was later used by the authors of [22] to develop a certain type of statistical model to support the decision to close a mine owing to air quality control.

An example of the use of mathematical and statistical techniques to predict copper recovery by leaching is given in [10], where the authors use Minitab^®^ software to create three statistical models: linear, quadratic, and cubic fit. A regression of the best subsets was conducted for each model, obtaining an R^2^ = 92.3% fit for the cubic fit model. On the other hand, machine learning methods are used in studies such as those above, which are further described in the section below.

Furthermore, there are many cases and discussions on how supervised learning methods (e.g., single-layer neural networks) and deep learning methods (e.g., convolutional neural networks) may be used in different kinds of analyses [23,24] and using different structures of data [25,26]. Next, a brief description of the supervised machine learning algorithms used in this work is given.

### 2.2. Artificial Neural Network

Artificial Neural Networks (ANNs) imitate the biological neural network function of the human brain. They can approach every continuous function showing structural formation and complexity [18,27]. In petrophysics, they are widely used for calculating petrophysical properties, generating synthetic records and classifying rock facies [28].

The basic elements of a neural network include layers, neurons, and activation functions. The neural network structure is simplified by using a feedback-based neural network with a hidden layer.

The neurons in two adjacent layers are connected by a weighted linear combination and an activation function to introduce non-linearity. The most common activation functions are sigmoid, ReLU, and TanH.

Owing to the dataset size herein, the activation function is limited to a sigmoid function for the hidden layer and a SoftMax function for the output layer. However, ReLU and TanH are preferred for deep neural networks because they do not have the vanishing gradient problem [29]. The value of each hidden layer is calculated as [30]:(1)$$y=\sigma \left(\overset{n}{\underset{i=1}{\sum }}w_{i}x_{i}\right)$$
where x_i_ are input records, w_i_ is the corresponding weight to x_i_, and σ is the sigmoid function defined as:(2)σx= 11+e−x

For each x_i_, the cost and gradient functions are calculated using backpropagation and the weights are optimized with a conjugated gradient solver.

### 2.3. Support Vector Machine

Support Vector Machine (SVM) is a classifier that defines a decision limit by optimally separating two classes. In linear cases, the decision limit becomes a hyperplane. The construction of the hyperplane separating two classes is based on a sample subset near the decision limit (support vectors). In the linear case, two classes are defined by limits, that is, wx-b ≥ 1 and wx-b ≤ −1. A margin is defined as the distance between these two hyperplanes, 2/|w|.

SVM training maximizes the margin between the training points of two different classes. A restriction-delimiting parameter C is used to control penalization when training instances are classified incorrectly. For a high value of C, the SVM tends to generate a smaller margin at the risk of overfit. A small value of C results in more erroneous classifications at the expense of training precision [2,12]. An SVM advantage is that a kernel trick is easy to use for mapping low-dimensional data and linear decision limits in a high-dimensional space to solve non-linear classification problems. Some of the indicators most frequently used are [30]:(3)$$K\left(x,\;x\prime \right)=\exp{\left(x^{T}\;x\text{′}+c\right)}^{d}$$

The Gaussian radial basis function (RBF) is defined as shown in Equation (4). For the RBF and to reduce the number of hyperparameters, only Gauss kernel functions are considered. The kernel is controlled by kernel scale y. A one-to-one classification is implemented in the process to improve the multiclass classification results because the computing cost does not significantly increase with the number of classes.
(4)$$K\left(x,\;x\text{′}\right)=\exp\left(-\text{y}\|\text{x}-x\|\right)$$

### 2.4. Random Forest

Random Forest (RF) is a supervised learning algorithm derived from decision trees (DT), which is frequently used for developing a predictive model [31,32]. As described in [3,31,33], DT is basically a hierarchical set of nodes (beginning with a root-node), where each node contains a decision based on the comparison between an attribute and a threshold value. DT-based learning starts with the observation of an object represented by the branches of a tree and ends with certain conclusions related to the target value of an object (represented by tree values).

The RF works in a labelled dataset (training set) to make predictions and produce a model. The resulting model can be later used to classify non-labelled data. The method combines the idea of bagging with the random selection of characteristics so as to construct decision trees with controlled variance [31]. One of RF main benefits as a model is that it can be used for determining the importance of the variables in a regression or classification problem intuitively [34,35]. This importance is calculated with a metric, according to the impurity decrease in each node used for data partitioning [36]. In the case of classification, the class determined corresponds to the mode of the classes provided by each tree.

## 3. Materials and Methods

### 3.1. Data-Driving Techniques in the Copper Industry and the Methodology

Currently, there are some cases in which data-driven algorithms are used in the industry of copper production by leaching to analyze data and provide results, such as models for predicting the quality of the copper recovered [3]. Based on data availability, data-driven techniques such as machine-learning algorithms (ANN, SVM, and RF) are implemented as copper recovery classifiers. The definitions of these algorithms and their main characteristics applicable in this context are described below.

In the field of copper recovery by leaching, there are a lot of parameters that have an influence on the resulting recovery; some of them can be controlled but other cannot be directly controlled [3]. This section describes the experimental parameters and the sizes of the used datasets.

The methodology used in this study is detailed graphically in Figure 2. Each stage of the methodology is described below.

Data preparation. For this first step data from the copper heap (called plant) and data from the laboratory (called lab) were used, both data sets were acquired in accordance with an experimental design, described below. The datasets contain the features of both processes (the leaching process on the plant and the leaching process in the lab). In both cases, data were prepared according to the results of the predictive variable measurements on the heap and in the lab. In this stage, .csv archives were used with data collected at the plant and in the lab. The class “copper recovery”, and associated variables are detailed in Table 1.Labels generation. The second stage consisted in generating labels for the dependent variable Y (copper recovery—see Table 1) in both datasets, according to the threshold values of the other predictive variables. This process was aiming to prepare the entry data for the data mining algorithms. This resulted in datasets with labels for the dependent variable Y.Model generation. The third stage consists in training for developing predictive models, using some parts of the datasets for training and the remaining ones for validation; the details are given in Section 4.Validation and analysis of results. The fourth stage deals with the validation and interpretation of the results. In this stage, comparative statistical tests using classifier performance measures were utilized to determine the quality of the models developed. The models that resulted from these tasks were analyzed and interpretated, according to expert knowledge and experience in the field of copper mining exploitation by leaching.Prediction. Using the validated models, predictions of copper recovery by leaching were made, and these were compared using the techniques described in the previous step.

### 3.2. Data Preparation and Experimental Design

For this study the experimental design is based on the previous models [3,10]. In the same way, the data used as input for the models were provided by the Franke copper mine. This mine is located in Atacama region, Chile. More details about this company and their mining processes can be found in [3,10]. One of the main goals of the leaching process is to obtain the greatest copper production by saving resources and being the least possible aggressive to the environment.

As has been described by the authors of [3], the copper leaching process involves tasks, such as irrigation beginning and maintenance, agglomerate condition evaluation, drainage distribution, pool solution inventory, and PLS flow evaluation. Figure 1 illustrates these processes. In later stages the process of distribution and deposition of the material leached at the plant (harvest) are made. At the Franke mine the leaching cycles are planned to last 65–100 days.

Two kinds of resulting data related to the material leached were considered for this work. The first one is related to a dynamic plant and the second one is related to a pilot plant (or piling plant). For each of them, a dataset composed by records used in [10] (2017) and a dataset composed by records used in [3] (2020) were considered.

In order to facilitate the interpretation and relationship with the input data, the following notations were used: A1 and B1 identify the datasets that were collected for the work reported in [10] (2017) and A2 and B2 identify the datasets that were used in the work reported in [3] (2020), the labels A1 and A2 identify the operational data and the labels B1 and B2 identify the laboratory-piling data (pilot data).

Operational data correspond to data that were collected during periods called leaching cycles beginning after soil piling and the starting of the irrigation process, since day one to the last day of production. Similarly, laboratory data corresponds to production that was conducted in two agglomerate tanks of the same dimensions with a material whose granulometry was less than 13 mm in diameter (more details in [3]).

In this work a total of 20,000 records from the operational plant and 15,000 records from piling were analyzed. Using these data, the following datasets were prepared: 4916 operational records corresponding to 30 leaching cycles during an average of 67 days were cleaned and prepared, and they are A1, 3772 operational records corresponding to 33 leaching cycles during an average of 61 days were cleaned and prepared, and they are A2. In addition, 3863 piling records, corresponding to 65 leaching cycles during an average of 61 days, were cleaned and prepared, and they are B1, and finally 3030 piling records, corresponding to 63 leaching cycles during an average of 63 days, were cleaned and prepared, and they are B2.

To develop the predictive models the variables shown in Table 1 were used. Each algorithm has its own characteristics and procedure. Therefore, the predictive variables had to be adjusted. Table 1 shows the initial discretization of the variables needed to use the algorithms.

The data from the operation plant were obtained at a frequency of 4 h for 1 year. For some periods, irrigation was stopped in some cells or modules in service. Due to these periods, inconsistent results were disregarded, and the corresponding data were not considered for the data collection process. Some “noise” in the system and useless information were disregarded as well. Regarding the pilot data, the collection method was the same as that used for the operational data.

Mono class granulometry (mm): ore particle size after the crushing process.Irrigation rates (Lhr/m^2^): amount of ILS solution added per m^2^ of the upper stockpile surface.Total acid added (g/L): concentration of acid added per liter of ILS.Stocked high (m): the height of the stockpile in operation.Total copper grade (%): the amount of copper in the ore to be leached.CO_3_ grade (%): the amount of carbonate in the mineral to be leached. This variable increases the acid consumption during the leaching process.Leaching ratio (m^3^/TMS): the amount of ILS added to the stockpile per amount of ore in the stockpile.Operation (days): the number of days during which the stockpile is in operation.Soluble copper grade stacked (%): in mineralogy, some ores can be leached, while others such as sulfates cannot. The amount of copper soluble in the ore is measured.Number of stockpiles: corresponding to the identification of the stockpile to be leached.Copper recovery (%): the copper recovered daily during the leaching process.

To generate the models, the free educational version of Rapidminer 9.7^®^ was used. This software tool allows developing models with desired characteristics and available datasets [35].

Before developing the models with the algorithms selected for this study (FR, SVM, ANN), three discrete values or labels were created for the dependent variable (Y). These labels were created from the continuous values of the variable Y described in Table 1, and also considering both the threshold values of the predictive variables (Table 1) and expert knowledge about the leaching process, available at the Franke Company, Aarburg, Switzerland.

These labels were modified considering earlier results, when the models were developed with RF. These models helped identify knowledge: when the value of variable X7 is lower than 4.1 and the value of X8 is greater than 40, copper recovery (Y) does not increase significantly as compared with the previous 10 days. Observing these earlier results and the interpretation of models with RF, new threshold values were determined for Y. Together with an expert from the Franke company, the threshold value of X7 = 4.2 was selected for creating the new labels for Y (details are shown in Section 4.1). These values were: “1” for optimal values of Y, when X7 ≤ 4.2 and, “0” for less-favorable values of Y when X7 > 4.2. For SVM, the same two ranges for Y were used in order to create two classes.

The variable X7 is a consumption and inventory indicator of the sulfuric acid used in the leaching irrigation process (variable X2) and allows establishing the indicators of the process cost. The relationship between variable X7 and operational costs is as follows: the smaller the variable X7, the lesser the operational costs. However, if the variable X7 decreases, it affects the copper recovery (dependent variable).

The relationship between the variables X and Y is neither bidirectional, nor linear, i.e., an increase in the value of the variable X7 does not necessarily result in higher copper recovery (Y). However, a value greater than 4 does not give evidence of a significant slope in copper recovery increase. But a value 4.2 for the variable X7 is considered as a threshold value to separate recovery and generate recovery ranges giving rise to the labels described below: “low” for a recovery below 45% and X7 in the interval (0.0012,4.2), and “high” for a recovery over 45% and X7 in the interval [4.2,15).

To further describe the abovementioned and clarify the “close relation” between the variable X7 and class Y, two “type” cycles are shown in Figure 3, highlighting recovery and also the leaching ratio corresponding to each of the cycles represented. Figure 3a shows two examples of the evolution of copper recovery (variable Y), i.e., two cycles. One of them takes little more than 100 days (cycle 1), while the other takes about 70 days (cycle 2). Figure 3b shows that the value of X7 for cycle 1 follows an upward trend until reaching values higher than 14.0 after about 100 days, while the values of X7 for cycle 2 remain close to the referential value of 4.1. Figure 3 shows that the values of Y for cycle 2 remain constant (between 40 and 57, approx.) from day 40, while the values of Y for cycle 1 early exceed the value of 60 (after about 20 days of the exploitation).

### 3.3. Validation Using Performance Measures

Once the models are developed, they must be validated through different techniques. Often, the models are not validated with the same data used for training the classifier. As reported by the authors of [37], the k-fold cross-validation method is a good alternative for validating methods as those developed in this study. In the k-fold cross-validation, the (simple) original training set is partitioned into k disjoint sets, one subset being retained for validation and the remaining groups (k-1) randomly selected to be used in training, while the only remaining group of samples is used for creating the cross-validation error function. This function is minimized by changing the values in each training of the automatic learning algorithm [9,16].

To avoid overfitting in this study, similarly to [15], the training and validation sets were generated independently for each algorithm, using each dataset. So, for training with each algorithm (RF, SVM, and ANN), each dataset (A1, B1, A2 and B2) was used, separating the data into a training set and a validation set, as described above.

There are several statistical tests using classifier performance measures which work for the same dataset. In addition, there are several alternative performance measures to compare the goodness of classifiers. Here, performance measures that may be applied in the three classification algorithms used were selected [3,9].

In order to carry out the validation, and based on the data obtained in the experiments, we will obtain a confusion matrix. This matrix facilitates the analysis needed to determine where classification errors occur. The confusion matrix is a table that shows the distribution of errors in the different categories. The values of merit necessary to evaluate the performance of the classifier to be implemented will be calculated using this matrix. The confusion matrix is a 2 × 2 matrix with numerical values a, b, c, and d, which are the result of the classified cases, where a is the sum of the true positive cases, b is the true negatives, c represents the false positives, and d corresponds to the false negatives [24].

Next, the measures of merit of each classifier from previous studies [3,9,10,33] were used in a similar way. The measures of merit used in this study help to determine the quality of the predictive models developed and are based on data from the confusion matrix and the result of training with each classification algorithm. These values of merit are the following:1.Accuracy (acc) corresponds to the ratio of correctly classified samples from all the samples in the dataset [38]. This indicator can be calculated with the confusion matrix data, according to Equation (5), assuming that the dataset is not empty.
(5)$$\mathrm{acc}=\left(a+d\right)/\left(a+b+c+d\right)$$

2.Precision (p) is the proportion of true positives (a) among the elements predicted as positive. Conceptually, precision refers to the dispersion of the set of values obtained from the repeated measurements of a quantity. Specifically, a high precision value (p) implies a low dispersion in measurements. This indicator can be calculated according to Equation (6), assuming a + b ≠ 0.

(6)$$p=\left(a/a+\;b\right)*\;100$$

3.Recall (r) is the proportion of true positives predicted among all elements classified as positive, that is, the fraction of relevant instances classified. Recall can be calculated according to Equation (7), assuming a + c ≠ 0.

(7)$$r=\left(a+d\right)/\left(a+c\right)$$

4.Matthew’s correlation coefficient (mcc) is an indicator relating the predicted versus the real values, creating a balance between the classes, considering the instances correctly and incorrectly classified into classes quite different in size and with a significant number of observations [39]. The mcc value can be calculated according to Equation (8), assuming that the destination dividend (τ) is not zero.
(8)$$\mathrm{mcc}=\left(\left(a+b\right)-\left(c+d\right)\right)/\tau$$
where
(9)$$\tau \;=\sqrt{(\left(a+d\right)*\left(a+c\right)*\left(b+d\right)*\left(b+c\right)}$$

In each of these models, the values of merit of Equations (5)–(8) were calculated, being useful for comparing the goodness of the models developed. These calculations, interpretations, and comparisons are described in the next section.

## 4. Results and Discussion

### 4.1. Results

This section describes the results obtained. As mentioned above, in order to facilitate their interpretation and relationship with the initial data, the following notation was used: dataset A1 is the stockpile dataset used in [10] and dataset A2 the stockpile dataset used in [3]. Analogously, dataset B1 is the pilot dataset used in [10] and dataset B2 the pilot dataset used in [3].

First, a training using RF was made. The resulting values of parameter “classification error” for each training were as follows: 0.0014 for training with dataset A1; 0.0169 for training with dataset A2; 0.0403 for training with dataset B1; and 0.0715 for training with dataset B2. The value of parameter “classification error” increases from training with the first dataset to the last one, but no relationship was found between this result and the possible interpretations of the model results.

Table 2 shows the RF total classification and the datasets for each copper recovery label (“low”, “medium”, “high”), corresponding to datasets A1 and A2, and Table 3 shows the corresponding information for datasets B1 and B2. In Table 2, the results show that classification precision is always over 95%. On the other hand, Table 3 shows that classification precision is over 99% for the three ranges. Additionally, for both tables the difference between the real and predicted value is lower than 30% for all classes.

This may be interpreted as a good approximation of the RF-generated models to what is actually observed in each dataset. It also shows that the classification precision is always over 97%, while the real and predicted value is below 30% for all the class cases.

Figure 4 shows the real vs. the predicted values in each dataset, according to the classification labels. The abovementioned is related to the interpretability of results from the RF-generated models. Figure 5 shows no significant changes between the values of merit in any of the four scenarios. These data, together with the interpretation of trees described in Figure 5, show that separating the datasets is not significant, concerning the three values of the labels in the class. Rather than considering the threshold value X7 = 4.1, it is possible to optimize the separation into two groups in the class as follows: “low” in the class variable for values X7 < 4.1 and “high”.

In detail, Figure 5a shows a result for dataset B1, clearly illustrating that “medium” and “high” groups may be regrouped into one group under the threshold X7 ≥ 4.097, while Figure 5b shows a result for dataset A2, illustrating that “medium” and “high” groups may be regrouped into one group under the threshold X7 ≥ 4.108.

With this information a preliminary analysis for the advisability of using these three labels for the variable Y (“high”, “medium” and “low”) was carried out by using the information from Table 2 and Table 3, and Figure 4 and Figure 5. First, using the datasets as input, the RF algorithm and the three labels for the variable Y, predictive models for Y were generated. Table 2 shows the classification results using A1 and A2 with 99.95% as final accuracy, similar results using B1 and B2 are shown in Table 3, where the final accuracy is 99.93%. Both records labeled as “medium” and “low” for Y can be regrouped in “low” because it is not exceeding the threshold value 4.2 for values of Y in the range 60–80% of recovery.

Figure 4 shows the real vs. the predicted value in each dataset, according to the classification labels, this information together with the interpretation of trees described in Figure 5. It shows that separating the datasets is not significant concerning the three values of the labels in the class. This is because Figure 5a shows that the limited value between “high” and “medium” is 4.097 and Figure 5b shows that the limited value is 4.1. Rather than considering the threshold value X7 = 4.2, it is possible to optimize the separation into two groups in the class as follows: “low” in the class variable for values X7 < 4.2 and “high” for all other cases. Based on the description above, the records labeled as “medium” and “low” for Y were regrouped in “low” and for the other records the label “high” is maintained. All other models (described below) were generated using this new two labels for Y (“low” and “high”).

After the training with RF and following the configuration described above, both the training with SVM and the training with ANN were made. For SVM training, the value of C was set at 0.0. For training with ANN and each dataset, an operator called feed-forward neural network from Rapidminer © was used, with training cycles = 200 and a value of learning rate = 0.01. Table 4 shows the values of merit for the classifications of these algorithms. These values were obtained from the classification values in the confusion matrices shown in Table 5, Table 6 and Table 7.

### 4.2. Discussion

In this study a total of twelve predictive models were generated, in detail: four predictive models using RF, four with SVM, and four using ANN. For these optimizations 80% of the data were used for cross-validation and 20% for validation, in a similar way to [3]. In addition, four measures of merit including acc, p, r, and mcc were used for each algorithm to evaluate the quality of all these machine learning methods.

Table 4 shows the values of merit for validating the quality of the model classification results. This table shows that the classification precision (class-precision) in all models is over 98.50%, while the ideal classification precision is 100%. The mean value of acc = 0.943 indicates that almost all the samples in the datasets were correctly classified, while the worst absolute value in all the models was obtained for dataset B1. In the same way, the mean value for p = 88.47 indicates a reasonable middle value for dispersion for all datasets, while the worst absolute value of p in all the models was obtained for dataset B2. This low value of p in the results of dataset B2 may be associated with high data dispersion in the data from the pilot dataset.

In Table 4 the result of the variable of merit mcc is also remarkable. The mean value is mcc = 0.232. Since all the values are low, the model prediction is quite close to the real value. Considering this result, it is then possible to interpret that the datasets and the preparation of the predictive variables and labels are correct. In relation to the parameter r (mean value of r = 0.995), no significant differences have been observed among the models. On the other hand, the well-classified figures (in the models’ confusion matrix tables) are above 95.98% as shown in Table 5, Table 6 and Table 7.

It is known that RF, SVM, and ANN are probabilistic models that provide a powerful formalism for representation, reasoning, and learning in the presence of uncertainty. Considering the results given in Table 4, ANNs show the best capability to reasoning over the datasets, while RF and SVM have similar behaviors in the data used in this work. An additional preliminary conclusion is that the values seem to be mostly balanced, without the classes bringing down the performance in a significant way. This is a very important feature for models when imbalanced data is used as input, as indicated in [38,40].

Table 5 shows the classification obtained with RF. In particular, it should be noted that the final accuracy is over 99.55%. The best classification obtained has been for the label “low”, with a total of 100% of precision for the 99.97% of the total of samples.

Similarly, Table 6 shows the classification obtained with SVM. The final accuracy is over 98.65%. This value is lower than the value obtained by using FR. The best classification obtained has been for the label “low”, with a total of 98.69% of precision for the 99.12% of the total of samples.

In the same way, Table 7 shows the classification obtained with ANN. The final accuracy is over 99.51%. This value is also above the value obtained by using SVM. Similarly, the best classification has been obtained for the label “low”, with a total of 99.41% of precision. All these Table 5, Table 6 and Table 7 are showing that the classification precision for all generated models is over 90.00%, while the gap between the real and predicted value is below 10%.

Figure 6 illustrates the contrast described above. As shown in Figure 6a, the classification closest to reality is obtained with ANN, representing 99.72% of the real data, while the worst behavior is obtained with RF, representing 88.91%. Figure 6b,c show that the best classification is for ANN, with 98.36% and 99.87%, respectively. The worst classification in Figure 6b is obtained with SVM (93.01%) and in Figure 6c with RF (70.01%). Figure 6d shows a change in tendency corresponding to dataset B2 since the best result was obtained with SVM (99.61% real data), although the worst result is obtained with RF, the same as in previous cases. In summary, this tendency was observed for the rest of the figures, where the best classification is for ANN and the worst is for RF except in Figure 6f where the worst corresponds to SVM. Figure 6e,h show the same tendency: the best classification is for ANN and the worst is for RF, except in Figure 6f where the worst is for SVM with dataset A2. This may be explained by the fact that dataset A2 corresponds to lab data in a more controlled environment as compared with the real conditions in a leaching stockpile.

Table 8 shows the detailed values of each classification for each dataset. It also shows a summarized value called CCP (classification certainty percentage), i.e., the certainty in the classification of each algorithm used in this study. The aforementioned and what is described in Table 8 is knowledge obtained from the models, causing a favorable impact on the work context since it can be used in future copper exploitation with similar characteristics.

Currently, machine learning methods are being developed for predicting, improving, and optimizing industrial processes as shown in [15] or [38]. These methods have caused a positive impact on process optimization to improve the plant production of various industries. Likewise, the authors of [3,10] report previous work, particularly in the leaching context, that contributes to the improvement of the interrelations between the variables measured, which influence the leaching process.

In the same line of thought, this study allows improving and optimizing the leaching process, whose results enable predicting stockpile behavior. Recent studies show the interest of mining companies to optimize their processes and planning to make a better use of resources.

The results of this study are somehow comparable to those of previous studies [12,14]. This is because they also find relations between elements of interest for their research in the mining context, using data-driven techniques such as ANN or SVM. However, unlike those studies, this study measures performance concerning the quality of prediction with the data-driven techniques used. This study is also similar to [38] because it uses artificial intelligence techniques such as SVM to identify relations between parameters, although they direct their study to the identification of close multidimensional relations, while this study does not consider this kind of relations between parameters.

The work described in [15] also shows an approach similar to this study but for prediction of organic solvent nanofiltration (OSN) in the building materials industry. The methodology followed in [15] is similar in some way to the method described in this manuscript. This is because the algorithms RF, SVM and ANN were used for the prediction of OSN performance.

An interesting study for comparison is reported in [41], which proposes a math-based mechanism for the leaching process and a mathematical model to identify and reduce the error in the mathematical modelling process of the hydrometallurgical leaching process. Unlike this study, they develop an ad-hoc mathematical model, while the models in this study could be used in other hydrometallurgical leaching processes, if there is a certain amount of experimental data with similar characteristics.

## 5. Conclusions

This study compares three supervised machine learning algorithms for classifying copper recovery quality prediction in a leaching process, using real data collected in a copper mine in the north of Chile. In summary, the three artificial intelligence methods—RF, SVM, and ANN—may be used to develop predictive models for copper recovery, identifying and validating the most influential predictive variables of class Y (copper recovery). This study uses datasets prepared and tested in previous studies to develop predictive models as detailed in [3,10]. One of the main contributions of this study is the use of four datasets: two related to real operational data and two related to laboratory-piling (pilot), used by developing predictive models with RF, SVM, and ANN.

The resulting models were validated using the cross-validation method, 80% data were used for cross-validation and 20% for validation. This allowed developing validated and reliable models with a minimal error value in trainings. This resulted in an average accuracy of over 95% in all the predictive copper recovery models using the four datasets prepared for this study and an average precision of over 98% in all the classifier trainings. In addition, measures of merit (acc, p, r, mcc) for each classifier were used in order to determine the quality of the predictive models developed. The models obtained show the following mean values: acc = 0.943, p = 88.47, r = 0.995, and mcc = 0.232. These values are highly competitive as compared with those obtained in similar studies using other approaches in the context.

For this work some interesting results with respect to RF, SVM, and ANN have been achieved. For example, the parameter optimization with SVM models is noticeable for handling non-linear data. This method was useful to draw decision boundaries between data points from different classes and separate them with maximum margin. The RF method was also useful for identifying the threshold values of the predictive variables that influence class values in the prediction. Another interesting conclusion is related to the closeness of predictions to real results.

Particularly in this case, the best method was ANN, confirming its potential for developing accurate machine learning models for copper mining exploitation and showing that the ANNs have the capability to tackle this non-linear problem in a better way than the other two algorithms. The above is not detrimental to the results of the other methods used (RF and SVM) since the results obtained with these two methods are also quite good and accurate.

Finally, as has been said throughout this paper, the use of artificial intelligence methods, in particular the use of three machine learning algorithms in conjunction with real data is not very common in the copper industry. Thus, we believe that the comparative study among these machine learning methods builds an excellent platform for future studies in this area.

The authors consider the followings topics for future research:Analyzing the system’s behavior with variables not currently measured in the field, such as stockpile permeability and mineralogy.Automatizing operational variables according to the predictive model to improve the planning of the leaching plant.

## Figures and Tables

**Figure 1 sensors-21-02119-f001:**
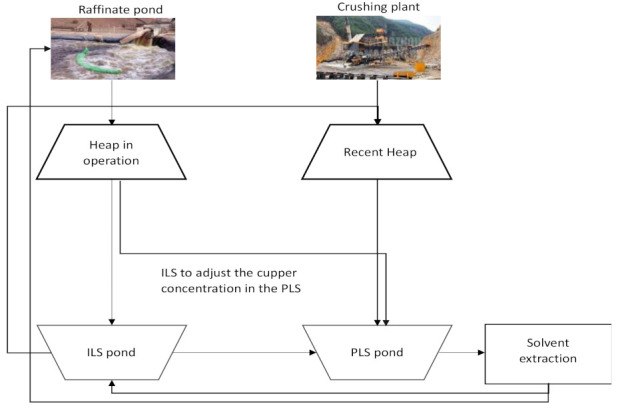
Main steps in the leaching process.

**Figure 2 sensors-21-02119-f002:**
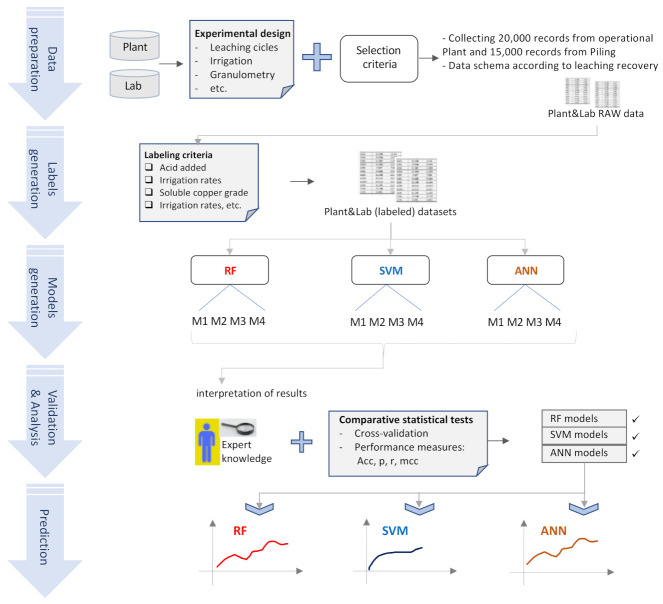
Workflow for copper recovery prediction using both datasets (Heap and Lab). This methodology consists of five steps, from data preparation to copper recovery prediction, using the model developed with Random Forest (RF), Support Vector Machine (SVM), and Artificial Neural Network (ANN) algorithms.

**Figure 3 sensors-21-02119-f003:**
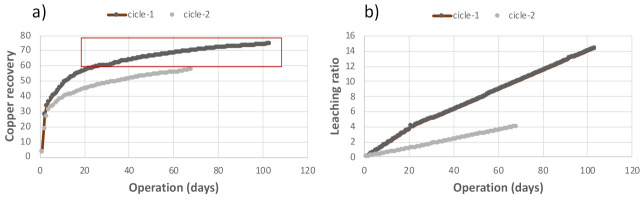
Examples of two copper recovery cycles and their relationship with variable X7. (**a**) shows levels of leaching ratio for cicle-1 and cicle-2, while (**b**) shows the different values of the variable Y influenced by X7 throughout each leaching cycle.

**Figure 4 sensors-21-02119-f004:**
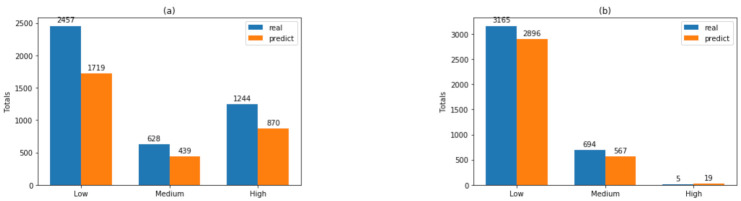
Real vs. predicted values with RF classifier for each dataset, using labels “low”, “medium”, and “high” for the dependent variables. The figure shows the total values of the real versus the classified values: (**a**) with dataset A1; (**b**) with dataset A2; (**c**) with dataset B1; and (**d**) with dataset B2.

**Figure 5 sensors-21-02119-f005:**
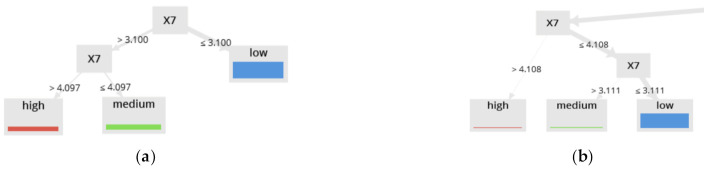
Two tree structures developed with RF, using two different datasets. (**a**) was generated using as input the dataset A1. (**b**) was generated with the dataset B2. The tree branches show the threshold values of variable X7 used, among other aspects, for defining the new labels of the dependent variables.

**Figure 6 sensors-21-02119-f006:**
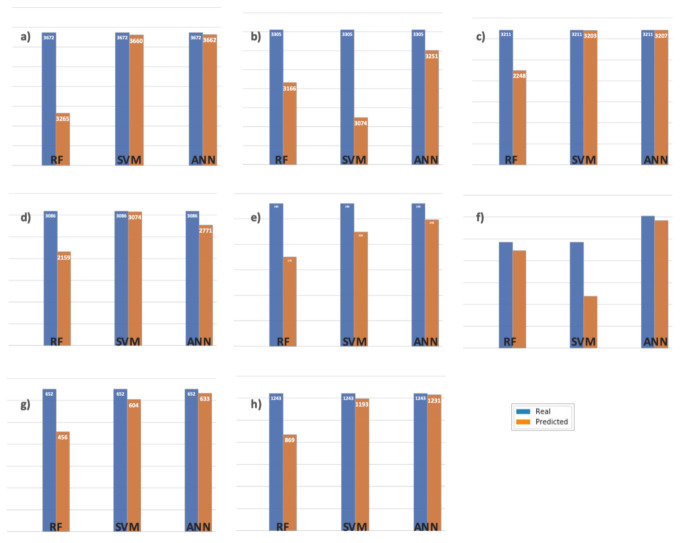
Comparison of real vs. predicted values for each dataset, considering the labels “low” and “high” for each dataset and each of the algorithms used (RF, SVM, and ANN). Data distribution: (**a**) “low” with dataset A1, (**b**) “low” with dataset A2, (**c**) “low” with dataset B1, (**d**) “low” with dataset B2, (**e**) “high” with dataset A1, (**f**) “high” with dataset A2, (**g**) “high” with dataset B1, and (**h**) “high” with dataset B2.

**Table 1 sensors-21-02119-t001:** Entry parameters of the statistical model. Variables X1–X10 are predictive, Y corresponds to the class or dependent variable.

Var	Description	Low	Normal	High
X1	mono class granulometry (mm)	[11.5,12)	[12,13)	[13,15)
X2	irrigation rates (Lhr/m^2^)	[6,8)	[8,11)	[11,14)
X3	total acid added (g/L)	[0.5,20)	[2050)	[50,75)
X4	stocked high (m)	[1,3)	[3,4)	[4,5)
X5	total copper grade (%)	[0.4,0.7)	[0.7,1.2)	[1.2,2.1)
X6	CO_3_ grade (%)	[0.5,4)	[4,6)	[6,10)
X7	leaching ratio (m^3^/TMS)	[0.012,5)	[5,10)	[10,15)
X8	operation (days)	[1,50)	[50,100)	[100,168]
X9	soluble copper grade stacked (%)	[55,65)	[65,80)	[80,98]
X10	Number of stockpiles	-	-	-
Y	Copper recovery (%)	[10,65)	[65,80)	(80,98]

**Table 2 sensors-21-02119-t002:** Confusion matrix with the classification generated with algorithm RF and datasets A1 and A2. A contrast between the classifications obtained with the datasets above is shown, along with accuracy, recall, and class-precision values for both matrices.

	Dataset A1 Accuracy: 99.97% ± 0.09%				Dataset A2 Accuracy: 99.94% ± 0.12%			
	true low	true medium	true high	class precision	True low	true medium	true high	class precision
pred. low	1719	0	0	100.00%	2896	0	0	100.00%
pred. medium	1	439	0	99.82%	0	567	0	100.00%
pred. high	0	1	870	95.00%	0	1	19	99.43%
class recall	99.97%	99.82%	100.00%		100.00%	99.73%	100.00%	

**Table 3 sensors-21-02119-t003:** Confusion matrix with the classification generated with algorithm RF and datasets B1 and B2. A contrast between the classifications obtained with the datasets above is shown, along with accuracy, recall, and class-precision values for both matrices.

	Dataset B1 Accuracy: 99.93% ± 0.16%				Dataset B2 Accuracy: 99.93% ± 0.14%			
	true low	true medium	true high	class precision	true low	true medium	true high	class precision
pred. low	1734	1	0	99.94%	1720	0	0	100.00%
pred. medium	1	511	0	99.80%	0	439	1	99.77%
pred. high	0	0	456	100.00%	0	1	870	99.89%
class recall	99.94%	99.80%	100.00%		100.00%	99.77%	99.89%	

**Table 4 sensors-21-02119-t004:** Values of merit for predictive models developed with algorithms FR, SVM, and ANN. These values of merit were obtained with datasets A1, A2, B1, and B2 and two values for class Y.

	Dataset	Class Precision	acc	p	r	mcc
RF	A1	99.700	0.9491	99.7558	1.0000	0.0563
	A2	100.000	0.9945	99.4548	1.0000	0.0529
	B1	100.000	0.8214	83.1261	1.0000	0.0026
	B2	99.850	0.9993	71.3012	1.0000	0.0016
SVM	A1	98.635	0.9491	99.7558	0.9955	0.0571
	A2	98.717	0.9945	99.4548	0.9878	0.0537
	B1	98.607	0.8314	83.1361	0.9877	0.0025
	B2	99.381	0.9993	71.3712	0.9878	0.0015
ANN	A1	99.381	0.9491	99.8350	1.0000	0.0042
	A2	99.921	0.9946	99.6177	0.9984	0.0437
	B1	99.381	0.8312	83.5156	0.9954	0.0018
	B2	99.381	0.9958	71.4451	0.9981	0.0011

**Table 5 sensors-21-02119-t005:** Confusion matrix with the classification generated using RF. This table shows the classification obtained with labels “low” and “high”, using datasets A1, A2, B1, and B2. Table 5 also shows the accuracy, recall, and class-precision values for both matrices.

	Dataset A1			Dataset A2		
	true low	true high	class precision	true low	true high	class precision
pred. low	3265	0	100.00%	3466	0	100.00%
pred. high	1	275	99.43%	0	1023	100.00%
class recall	99.97%	100.00%		100.00%	100.00%	
	**Dataset B1**			**Dataset B2**		
	true low	true high	class precision	true low	true high	class precision
pred. low	2248	0	100.00%	2159	1	99.95%
pred. high	0	456	100.00%	1	869	99.89%
class recall	100.00%	100.00%		99.95%	99.89%	

**Table 6 sensors-21-02119-t006:** Confusion matrix with the classification generated using SVM. This table shows the classification obtained with labels “low” and “high”, using datasets A1, A2, B1, and B2. Table 6 also shows the accuracy, recall, and class-precision values for both matrices.

	Dataset A1			Dataset A2		
	true low	true high	class precision	true low	true high	class precision
pred. low	3660	26	99.45%	3074	50	98.40%
pred. high	5	224	97.82%	12	894	99.00%
class recall	99.89%	90.60%		99.61%	95.98%	
	**Dataset B1**			**Dataset B2**		
	true low	true high	class precision	true low	true high	class precision
pred. low	3203	48	98.52%	2074	50	98.40%
pred. high	8	604	98.69%	12	894	99.00%
class recall	99.75%	92.64%		99.61%	95.98%	

**Table 7 sensors-21-02119-t007:** Confusion matrix with the classification generated using ANN. This table shows the classification obtained with labels “low” and “high”, using datasets A1, A2, B1, and B2. Table 7 also shows the accuracy, recall, and class-precision values for both matrices.

	**Dataset A1**			**Dataset A2**		
	true low	true high	class precision	true low	true high	class precision
pred. low	3662	2	99.96%	4951	2	99.84%
pred. high	3	248	98.80%	0	1092	100.00%
class recall	99.94%	99.20%		100%	98.42%	
	**Dataset B1**			**Dataset B2**		
	true low	true high	class precision	true low	true high	class precision
pred. low	3207	19	99.41%	2771	9	99.61%
pred. high	4	633	99.37%	16	1231	99.51%
class recall	99.88%	97.09%		99.81%	99.03%	

**Table 8 sensors-21-02119-t008:** Summary of the classification percentages for each dataset and class labels; CCP value for each classification algorithm, expressed as a percentage (%).

	A1	A2	B1	B2	A1	A2	B1	B2	PCC
	‘low’				‘high’				
RF	88.92	95.79	70.01	69.96	62.50	98.18	69.94	69.91	78.15
SVM	99.67	93.01	92.64	99.61	80.00	88.20	92.64	95.98	93.61
ANN	99.73	98.37	97.09	89.79	88.57	99.09	97.09	99.03	96.44

## Data Availability

All raw data remains the property of the company that allowed this study. The input data (anonymized data) used to support the endings of this study could be available from the correspondent author’s email with appropriate justification.

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
