# Peer review of "A Comparative Study on Supervised Machine Learning Algorithms for Copper Recovery Quality Prediction in a Leaching Process"

_sensors, 2021, doi:10.3390/s21062119_

Round 1

Reviewer 1 Report

The manuscript by Flores and Leiva has some merits, they used 3 machine learning algorithms for quality prediction in the copper mining industry. The work has practical relevance but does not show innovation from the machine learning aspect. There are some major and minor issues to be addressed prior to further consideration by the journal.

1, The authors should briefly justify why Artificial Neural Network, Support Vector Machine, and Random Forest, among all the available AI algorithms, were selected for the work.

2, An explanation should be provided why principal component analysis (PCA) was not performed?

3, The overfitting problem should be mentioned and details given with relation to the R values of training and test dataset, and the validation dataset, which reduces the likelihood of overfitting in machine learning methods. Therefore, the details of the validation dataset to remove overfitting problems should be explained in more details.

4, Add a section about the advantages, disadvantages and limitations of the proposed methods and then compare the proposed methodologies and results with the literature.

5, For general interest the authors should briefly mention the widespread emerging application areas of machine learning (polymers 10.1021/acsapm.0c00921; materials 10.1039/D0GC02956D; recycling 10.1021/acssuschemeng.0c06978; separations 10.1016/j.memsci.2020.118513).

6, Some of the figure and table captions are too short and vague, add more information so that the figures stand on their own, and facilitate the understanding of the manuscript.

7, The novelty and potential impact of the work needs to be emphasized.

Author Response

Reviewer 1
  comments reply to comments
1 1, The authors should briefly justify why Artificial Neural Network, Support Vector Machine, and Random Forest, among all the available AI algorithms, were selected for the work. A comparison of models was made in Section 1.1
2  The overfitting problem should be mentioned and details are given in relation to the R values of the training and test dataset, and the validation dataset, which reduces the likelihood of overfitting in machine learning methods. Therefore, the details of the validation dataset to remove overfitting problems should be explained in more detail. A paragraph was added in section 3.3, 2nd  par., to clarify how the problem of overfitting was added
3 Add a section about the advantages, disadvantages, and limitations of the proposed methods and then compare the proposed methodologies and results with the literature. Section 1.1 was added to highlight advantages, disadvantages, and general methodological steps to develop predictive models.
4 5, For general interest the authors should briefly mention the widespread emerging application areas of machine learning (polymers 10.1021/acsapm.0c00921; materials 10.1039/D0GC02956D; recycling 10.1021/acssuschemeng.0c06978; separations 10.1016/j.memsci.2020.118513). The papers suggested were read. They were helpful for improving several aspects of the manuscript. A number of them were included in the bibliography and the text. For further detail, they were used in the last part of Section 4 as a discussion of our results compared with results from previous studies. One of them was included in subsection 3.3 to support the definition of one of the quality metrics used in our paper.
5 Some of the figure and table captions are too short and vague, add more information so that the figures stand on their own, and facilitate the understanding of the manuscript. Descriptions were added in the captions of all the figures and tables to improve and facilitate the understanding of the data they include.
6 The novelty and potential impact of the work needs to be emphasized. This is a novel study on using these machine learning techniques to identify the characteristics and relations between the concepts of the leaching process, which may be influential for a more productive industry. Also, the comparison is novel because the literature does not contain previous studies such as the one here, fully focusing on copper leaching. These issues are highlighted in several parts of the document: two paragraphs before the newly incorporated Table 8; two paragraphs after Table 8; and the last part of Section 5, in the Conclusions, the last two bullets.

Reviewer 2 Report

Draw a flowchart from your work flow that briefly shows the process and in the Discussion section. Compare your results with the results of other researchers. In the final conclusion, make a new justification for your research. The introduction should be strengthened and newer sources should be used

Figure 2 is totally not clear!

Figure 3 is not clear, please draw it in better form.

Figure 6 requires more explanation.

Discussion part should be compared by other studies.

Author Response

Reviewer 2
  comments reply to comments
1.1 Draw a flowchart from your work flow that briefly shows the process and in the Discussion section.   Drawing a flowchart is complex and may simplify the results since a total of 12 models was developed (first, 4 with 3 labels for the dependent variable and the rest with 2 classes for the dependent variable). We agree with your observation and understand that it is necessary to describe the process fast. To respond to this observation, we restructured and added content. So, the content in Section 4.1 where results are described was separated and also the content in Section 4.2,  containing the conclusions of the study. Section 4.2 contains the descriptions of the captions in tables 2-7, while another table (8), summarizing the prediction percentages was added. Caption descriptions of tables 4-6 were also added. An in-depth data analysis of Figure 6 was added in the two par. after the figure (from line 625 on).
1.2 Compare your results with the results of other researchers.  A comparison with recent papers, concerning approach, study context, and results, was added in Section 4 (from line 661 on)
1.3 In the final conclusion, make a new justification for your research.  A justification of the future lines of the study was added at the end of Section 5 (from line 712)
1.4 The introduction should be strengthened and newer sources should be used The Introduction was reorganized to form a better approach to the aspects of interest. The content was included in both the Introduction and throughout the document, based on more recent papers. At the end of Section 4, references from this year were included for comparing similar papers.  
2 Figure 2 is totally not clear! Figure 2 was redesigned and text was added to it to facilitate its understanding both in the main body of the document and the caption of the figure itself. 
3 Figure 3 is not clear, please draw it in better form. Figure 3 was redesigned and the caption description was improved to facilitate understanding the information it contains. A par. was added before Figure 3 to improve its interpretation
4 Figure 6 requires more explanation. Figure 6 was redesigned and an extensive discussion of the data it contains was added (from line 625 on). Even, Table 8 was designed with derived data to complement the figure. Also, descriptive text, along with the interpretation of the data in this table 8 was added. 
5 Discussion part should be compared by other studies. A Discussion section (4.2) was included. A discussion concerning previous work was added in two paragraphs of Section 4.2 (from line 661 on)

Reviewer 3 Report

Suggestions included in the text.

Author Response

Reviewer 3
1 All comments that were written directly in the .PDF document were considered and the changes were made, for which I thank you.

Round 2

Reviewer 1 Report

The authors have addressed the comments.

Author Response

Response to Reviewer 1 Comments

Point 1: The authors should briefly justify why Artificial Neural Network, Support Vector Machine, and Random Forest, among all the available AI algorithms, were selected for the work.

Response 1: A comparison of models was made in Section 1.1

Point 2: The overfitting problem should be mentioned and details given with relation to the R values of training and test dataset, and the validation dataset, which reduces the likelihood of overfitting in machine learning methods. Therefore, the details of the validation dataset to remove overfitting problems should be explained in more details.

Response 2: A paragraph was added in section 3.3, 2nd par., to clarify how the problem of overfitting was considered

Point 3: Add a section about the advantages, disadvantages and limitations of the proposed methods and then compare the proposed methodologies and results with the literature.

Response 3: Section 1.1 was added to highlight advantages, disadvantages and general mythological steps to develop predictive models. A comparative of results with the literature has been added (three paragraphs after Table 8; and the last part of Section 5)

Point 4: For general interest the authors should briefly mention the widespread emerging application areas of machine learning (polymers 10.1021/acsapm.0c00921; materials 10.1039/D0GC02956D; recycling 10.1021/acssuschemeng.0c06978; separations 10.1016/j.memsci.2020.118513).

Response 4: The papers suggested were read. They were helpful for improving several aspects of the manuscript. A number of them were included in the bibliography and the text. For further detail, they were used in the last part of Section 4 as a discussion of our results compared with results from previous studies. One of them was included in subsection 3.3 to support the definition of one of the quality metrics used in our paper.

Point 5: Some of the figure and table captions are too short and vague, add more information so that the figures stand on their own, and facilitate the understanding of the manuscript. 

Response 5: Descriptions were added in the captions of all the figures and tables to improve and facilitate the understanding of the data they include.

Point 6: The novelty and potential impact of the work needs to be emphasized.

Response 6: This is a novel study on using these machine learning techniques to identify the characteristics and relations between the concepts of the leaching process, which may be influential for a more productive industry. Also, the comparison is novel because the literature does not contain previous studies such as the one here, fully focusing on copper leaching. These issues are highlighted in several parts of the document: two paragraphs before the newly incorporated Table 8; two paragraphs after Table 8; and the last part of Section 5, in the Conclusions, the last two bullets.

Reviewer 2 Report

The authors improved manuscript accordingly and this manuscript is acceptable.

Author Response

Response to Reviewer 2 Comments

Point 1.1: Draw a flowchart from your work flow that briefly shows the process and in the Discussion section.

Response 1.1: Drawing a flowchart is complex and may simplify the results since a total of 12 models was developed (first, 4 with 3 labels for the dependent variable and the rest with 2 classes for the dependent variable). We agree with your observation and understand that it is necessary to describe the process fast. To respond to this observation, we restructured and added content.

So, the content in Section 4.1 where results are described was separated and also the content in Section 4.2, containing the conclusions of the study.

Section 4.2 contains the descriptions of the captions in tables 4-7, while another table (8), summarizing the prediction percentages was added. Caption descriptions of tables 4-7 were also added. An in-depth data analysis of Figure 6 was added in the two par. after the figure (from line 611 on).

Point 1.2: Compare your results with the results of other researchers.

Response 1.2: A comparison with recent papers, concerning approach, study context, and results, was added in Section 4.2 (from line 650 on)

Point 1.3: In the final conclusion, make a new justification for your research.

Response 1.3: A justification of the future lines-of-study was added at the end of Section 5 (from line 712)

Point 1.4: The introduction should be strengthened and newer sources should be used

Response 1.4: The Introduction was reorganized to for a better approach of the aspects of interest, and descriptions of papers from this year were included. Content was included in both the Introduction and throughout the document, based on more recent papers. At the Introduction and the end of Section 4, references from this year were included for comparing similar papers. 

Point 2: Figure 2 is totally not clear!

Response 2: Figure 2 was redesigned. In order to facilitate the understanding, some text was added both in the main body of the document and the caption of the figure itself.

Point 3: Figure 3 is not clear, please draw it in better form.

Response 3:  Figure 3 was redesigned, and the caption description was improved to facilitate understanding the information it contains. A par. was added before Figure 3 to improve its interpretation.

Point 4: Figure 6 requires more explanation.

Response 4:  Figure 6 was redesigned and an extensive discussion of the data it contains was added (from line 612 on). Even, Table 8 was designed with derived data to complement the figure. Also, descriptive text, along with the interpretation of the data in this table 8 was added.

Point 5: Discussion part should be compared by other studies.

Response 5:  A Discussion section (4.2) was included. A discussion concerning previous work was added in two paragraphs of Section 4.2 (from line 640 on)